https://doi.org/10.1038/s41467-019-10921-7　　**OPEN**

# Direct observation and impact of co-segregated atoms in magnesium having multiple alloying elements

Xiaojun Zhao[1], Houwen Chen[1,2], Nick Wilson[3], Qing Liu[1] & Jian-Feng Nie[1,2,4]

Modern engineering alloys contain multiple alloying elements, but their direct observation when segregated at the atomic scale is challenging because segregation is susceptible to electron beam damage. This is very severe for magnesium alloys, especially when solute atoms segregate to form single atomic columns. Here we show that we can image segregation in magnesium alloys with atomic-resolution X-ray dispersive spectroscopy at a much lower electron voltage. We report a co-segregation pattern at twin boundaries in a magnesium alloy with both larger and smaller solutes forming alternating columns that fully occupy the twin boundary, in contrast to previous observations of half occupancy where mixed-solute columns alternate with magnesium. We further show that the solute co-segregation affects the twin migration mechanism and increases the twin boundary pinning. Our work demonstrates that the atomic-scale analysis of the structure and chemistry of solute segregation in metallic alloys with complex compositions is now possible.

[1] Joint International Laboratory for Light Alloys (MOE), College of Materials Science and Engineering, Chongqing University, Chongqing 400044, P.R. China. [2] Electron Microscopy Center, Chongqing University, Chongqing 400044, P.R. China. [3] CSIRO Mineral Resources, Clayton, VIC 3168, Australia. [4] Department of Materials Science and Engineering, Monash University, Melbourne, VIC 3800, Australia. Correspondence and requests for materials should be addressed to J.-F.N. (email: jianfeng.nie@monash.edu)

Grain boundaries play a critical role in controlling mechanical properties of many polycrystalline engineering materials such as lightweight magnesium alloys[1–5]. A major barrier to the wider use of magnesium products in aerospace and automotive industries is the control of deformation modes during thermomechanical processes and applications. Since the formability, deformation behaviour, and tension–compression yield-strength asymmetry of wrought magnesium products are all closely related to deformation twinning, there have been tremendous interest in gaining a fundamental understanding of the factors that dictate the development of such twins under different deformation conditions[6–9]. It is now known that the addition of RE elements to magnesium alloys can significantly weaken the recrystallization texture, and that a combined addition of RE and non-RE elements may generate an even weaker recrystallization texture than the single addition of RE[10,11]. The RE additions lead to more deformation twins that provide more nucleation sites for recrystallization grains having random orientations. It has been reported recently[12,13] that the combination of larger and smaller atoms of appropriate alloying elements can lead to much weaker texture and better formability by maximizing the co-segregation of these texture-controlling elements in grain boundaries. However, gaining fundamental insights from experimental observations of the effects of alloying elements on deformation twinning in Mg and more broadly in engineering materials has proved elusive[14–16]. Specifically, we need atomic-scale experimental evidence and an understanding of the structure and chemistry of twin boundaries in alloys that often contain multiple alloying elements to meet the industry application requirements.

Transmission electron microscopy is a powerful tool for this purpose. With a scanning transmission electron microscope (STEM) equipped with the spherical aberration corrector, it is nowadays possible to directly observe the distribution of relative heavy atoms using a Z-contrast based imaging technique such as high-angle annular dark-field (HAADF) imaging technique[3,17–20], or of lighter atoms such as oxygen, lithium or hydrogen in an annular bright-field image[21]. However, the analyses of such Z-contrast images become problematic when the alloys have multiple alloying elements. The chemistry of grain boundaries has been studied by atom probe tomography[2,22,23], but it is difficult to determine the detailed solute atom arrangement in the boundary. It is nowadays possible to obtain atomic-resolution energy-dispersive X-ray spectroscopy (EDS) maps in oxides, semiconductors and magnetic alloys[24–29], which are all resistant to electron beam damage. However, direct observation and identification of alloying elements segregated at the atomic scale is challenging for light alloys such as magnesium, aluminium and titanium, because the segregation in these materials is prone to electron beam damage. The beam damage is most severe for magnesium alloys and is particularly an issue when the segregated solute atoms become a single atomic column.

In this work, we demonstrate that it is possible to solve this difficulty by using atomic-resolution EDS at a much lower electron voltage. With this opportunity, we discovered a pattern of solute co-segregation in twin boundary, and a solute-segregation-induced mechanism of twin boundary migration. The alloy system that we selected in this work, Mg-RE-Ag (where RE represents rare-earth), forms the base of a group of magnesium alloys that have superior mechanical properties at both ambient and elevated temperatures[30,31]. The alloy contains Nd and Ag. Nd has a larger atomic size than Mg, Ag has a smaller size than Mg. Having higher atomic numbers on the periodic table, Nd and Ag themselves would have been unsuitable for Z-contrast imaging. Their distribution at the atomic scale can be revealed only by EDS.

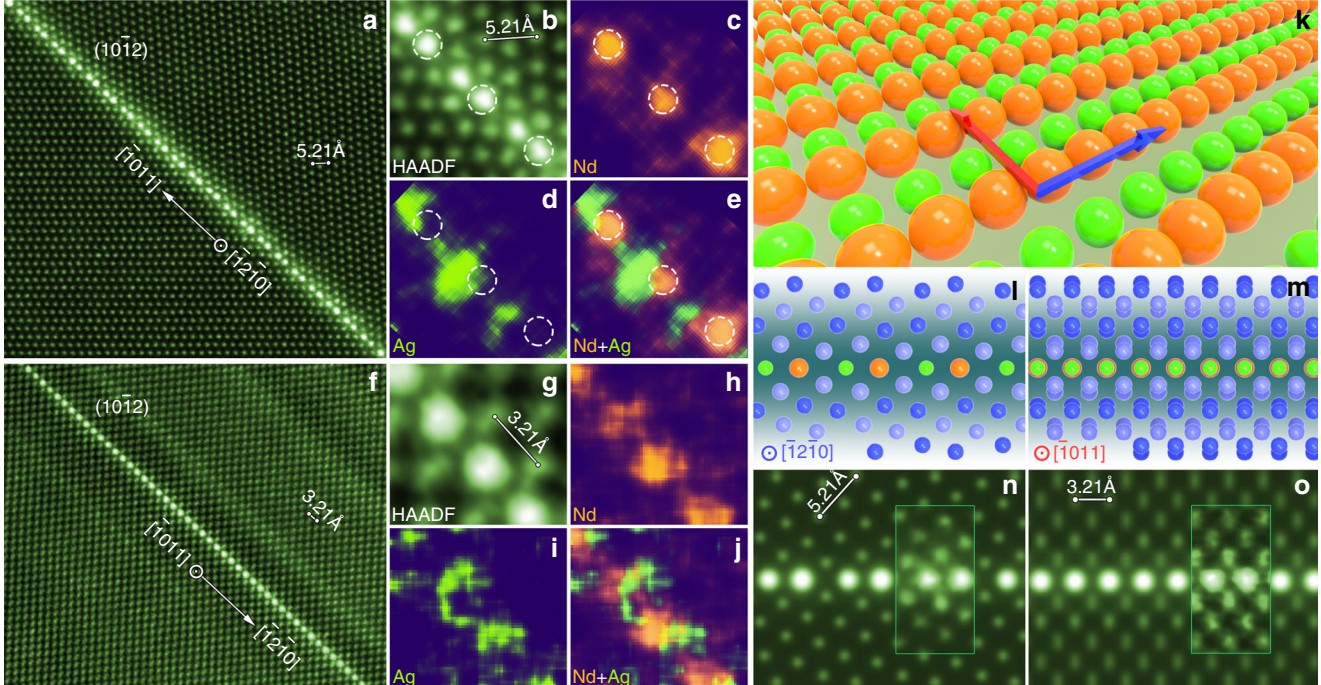

**Fig. 1** Alternating distribution of Nd and Ag columns in a (10$\bar{1}$2) twin boundary. **a** Atomic-resolution [$\bar{1}$2$\bar{1}$0] HAADF-STEM image. **b** Enlargement of a region in **a**. **c–e** Corresponding atomic-resolution EDS maps showing atomic columns rich in **c** Nd, **d** Ag and **e** (Nd + Ag). Dashed circles in (**b–e**) indicate extension sites. **f** Atomic-resolution [$\bar{1}$011] HAADF-STEM image. **g** Enlarged image of a boundary segment in (**f**). **h–j** Corresponding atomic-resolution EDS maps showing atomic columns rich in **h** Nd, **i** Ag and **j** (Nd + Ag). **k** Schematic diagram showing arrangement of Nd and Ag atoms within a (10$\bar{1}$2) twin boundary. Blue and red arrows indicate [$\bar{1}$2$\bar{1}$0] and [$\bar{1}$011] directions, respectively. **l, m** Segregation layer viewed along **l** [$\bar{1}$2$\bar{1}$0] and **m** [$\bar{1}$011]. **n, o** Simulated [$\bar{1}$2$\bar{1}$0] and [$\bar{1}$011] HAADF-STEM images, respectively. Insets in (**n**) and (**o**) are experimental images

## Results

**Co-segregated solutes in $(10\bar{1}2)$ twin boundaries.** Figure 1a–b shows $[\bar{1}2\bar{1}0]$ HAADF-STEM images of a $(10\bar{1}2)$ twin boundary in a plastically deformed and annealed sample. All atomic columns within this boundary exhibit brighter contrast than those in the matrix or twin. Because the brightness of an individual atomic column in a HAADF-STEM image is approximately proportional to the square of average atomic number, the brighter contrast indicates an enrichment in solute. Since the atomic number of Nd (60) and Ag (47) are both higher than Mg (12), it is difficult to identify whether these individual bright columns are rich in Nd, Ag, or both, based solely on the contrast of the HAADF-STEM image. Therefore, we tried atomic-resolution EDS mapping. Figure 1c–e provides atomic-resolution Nd, Ag and $(Nd + Ag)$ EDS maps of the twin boundary shown in Fig. 1b, successfully obtained at a much lower accelerating voltage of electrons, i.e. 120 kV instead of 300 kV that has been commonly used. These EDS maps reveal unambiguously that Nd atoms segregate exclusively to the extension sites (dashed circles in Fig. 1b–e) while Ag atoms occupy only the compression sites. This segregation pattern is different from that in Mg-Gd-Zn alloys where larger and smaller solute atoms segregate into only the extension sites[3]. We found that the Nd-rich columns are more stable than Ag-rich columns under continuous electron radiation, Supplementary Fig. 1. This is why the quality of the Nd EDS map is better than that for Ag.

To establish the arrangement of co-segregated solute atoms, the segregated $(10\bar{1}2)$ twin boundaries are also examined along $[\bar{1}011]$, which is the twining direction and is perpendicular to $[\bar{1}2\bar{1}0]$ and the normal of the $(10\bar{1}2)$ plane. When viewed along $[\bar{1}011]$, the twin and matrix exhibit identical projections of atomic columns, and the diffraction patterns of these two crystals are also identical, making it difficult to see the twin boundary at the atomic scale. However, the segregation of the solute atoms allows the twin boundary to be observed directly in HAADF-STEM images, Fig. 1f–g. All columns in the twin boundary exhibit brighter contrast, indicating an enrichment of solute along the viewing direction. While it is again difficult to distinguish Nd and Ag in the HAADF-STEM images, Fig. 1f–g, the corresponding atomic-resolution EDS maps indicate clearly that each atomic column contains both Nd and Ag atoms, Fig. 1h–j.

Based on the HAADF-STEM images and atomic-resolution EDS maps obtained from above two orthogonal directions, the distribution of Nd and Ag atoms within a $(10\bar{1}2)$ twin boundary was established and is shown schematically in Fig. 1k. Along the $[\bar{1}2\bar{1}0]$ direction (blue arrow), each atomic column contains either Nd or Ag atoms. When viewed along the $[\bar{1}011]$ direction (red arrow), the Nd and Ag atoms distribute alternately in each column. Figure 1l, m shows schematically projections of the segregation layer along $[\bar{1}2\bar{1}0]$ and $[\bar{1}011]$. Simulated images, Fig. 1n–o, shows good agreement with the experimental images.

**Co-segregated solutes in $(10\bar{1}1)$ twin boundaries.** The co-segregation phenomenon was also observed in $(10\bar{1}1)$ twin boundaries. Figure 2a shows a HAADF-STEM image of a $(10\bar{1}1)$ twin boundary in a deformed and annealed sample. Again, both extension sites (dashed circles) and compression sites are occupied by solute. Enlarged HAADF-STEM image and corresponding atomic-resolution EDS maps, Fig. 2b–e, indicate that Nd atoms segregate into the extension sites and that Ag atoms to the compression sites, forming a segregation pattern similar to that with $(10\bar{1}2)$ twin boundaries.

**Energies for solutes at twin boundaries.** To reveal the origin of the unusual co-segregation pattern, alternating columns of larger

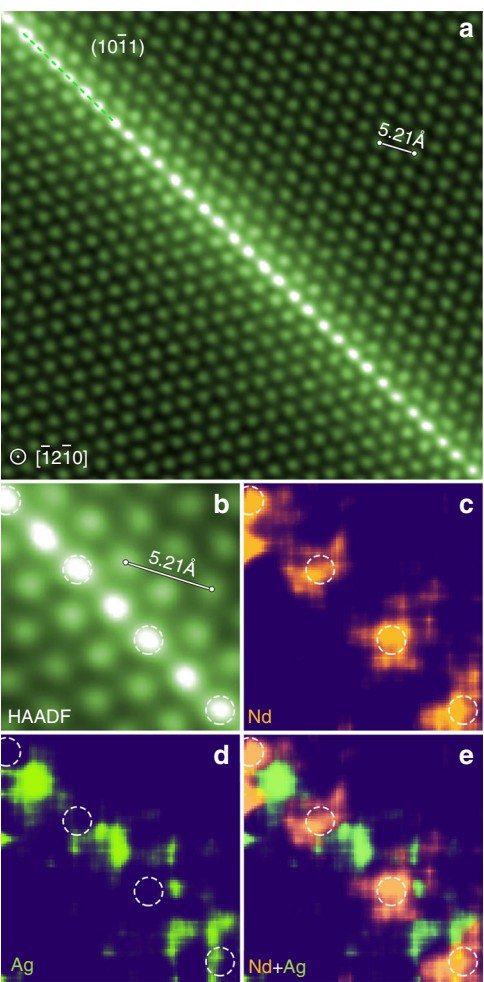

**Fig. 2** Alternating distribution of Nd and Ag atoms in a $(10\bar{1}1)$ twin boundary. **a** Atomic-resolution HAADF-STEM image viewed along $[\bar{1}2\bar{1}0]$. **b** Enlarged image of a region in (**a**). **c–e** Corresponding atomic-resolution EDS maps showing atomic columns rich in **c** Nd, **d** Ag and **e** (Nd + Ag). Dashed circles in (**b–e**) indicate extension sites

and smaller solute atoms occupying the entire twin boundary, we performed first-principles computations. The calculated relative energies are shown in Fig. 3 for a range of solute incorporations at the $(10\bar{1}2)$ and $(10\bar{1}1)$ twin boundaries. For the $(10\bar{1}2)$ boundary, it can be seen that for the extension site, the most favourable arrangement is for a fully occupied column of Nd in the $[\bar{1}2\bar{1}0]$ direction (Fig. 3a). In contrast to the previous observations[3] that both larger and smaller solute atoms co-segregate into only the extension site, in this case the presence of mixed atoms of Nd and Ag in a single column of the extension site leads to higher energy. The energy is further lowered if the compression site is fully occupied with Ag atoms, dashed line in Fig. 3a, which is in agreement with the experimental results. Figure 3b considers the energetics of the compression site and similarly finds the lowest energy when the compression site is fully occupied by Ag and the expansion site by Nd. For the $(10\bar{1}1)$ twin boundary full occupation of the extension site by Nd and the compression site by Ag is found to be most favorable (Fig. 3c–d), while for the expansion site a mix of Nd and Mg in the expansion site is found to be of similar energy to fully occupying the expansion site with Nd. Mixed Ag and Nd filling of the expansion site is less favoured. We have also performed calculations placing the Nd and Ag atoms one and two $(10\bar{1}2)$ layers away from the twin boundary. At one $(10\bar{1}2)$ layer away from the boundary there is a site larger than

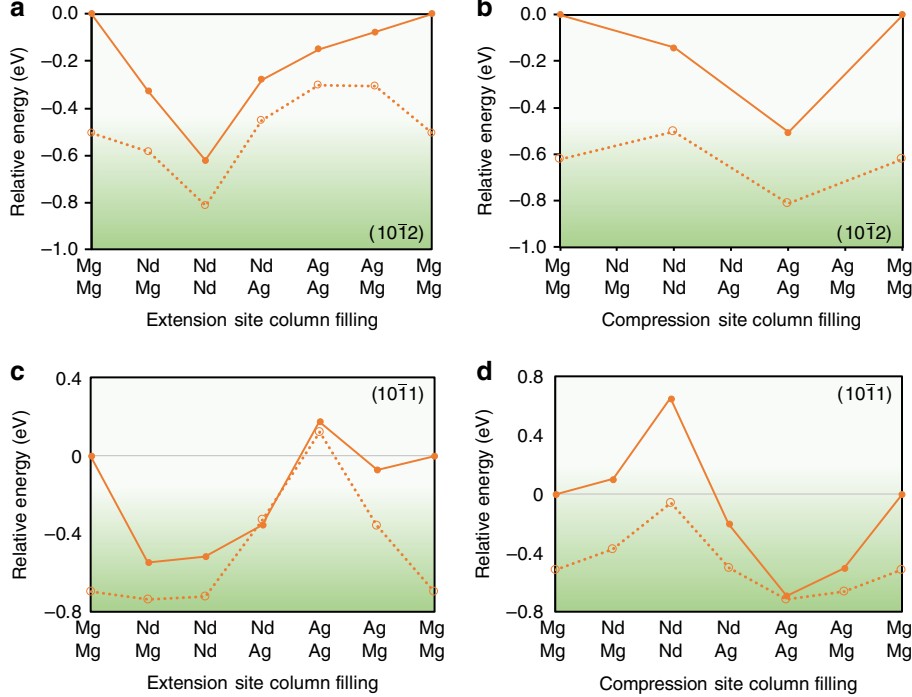

**Fig. 3** First-principles calculated energies for solutes at twin boundaries. **a** Relative energies of filling the extension site column in the ($10\bar{1}2$) twin interface, **b** the compression site column in the ($10\bar{1}2$) twin interface, **c** the extension site column in the ($10\bar{1}1$) twin interface, and **d** the compression site column in the ($10\bar{1}1$) twin interface. In (**a**, **c**) solid lines represent energies when the compression site is filled with Mg, while the dashed line gives energies where the compression site is filled with Ag. The solid line in (**b**, **d**) is for calculations with Mg filling the extension site and the dashed line has Nd filling the extension site. All energies are relative to complete filling by Mg in both the extension and compression sites

that found in the matrix, but not as large as the extension site in the boundary, and consequently, populating the first ($10\bar{1}2$) layer away from the boundary with columns of Nd and Ag is 430 meV less favorable than their placement at the boundary. In the second ($10\bar{1}2$) layer away from the boundary, site geometries are close to that found in the matrix and populating this second layer with Nd and Ag columns is even less favourable, being 610 meV higher in energy than when they are positioned at the twin boundary.

**Influence on twin boundary migration mechanism**. Effects of the co-segregation of Nd and Ag atoms on twin boundary migration mechanisms were examined by first-principles computation (Fig. 4). It is interesting to note that the presence of Nd and Ag in the twin boundary causes a change in boundary migration mechanism from the commonly accepted mode to a new mode. Atoms within the twin boundary plane and its first and second nearest neighbour plane behave differently when an external shear strain is applied. In the situation where there is no solute segregation (Fig. 4a), the angle associated with the original twin boundary ($\alpha$) decreases gradually with an increase in the shear strain, due to the shuffling of the compression site (C) Mg atoms moving in the opposite direction to the expansion site (E) Mg atoms, from 180° towards 164° (which is the stress-free angle of three neighbouring Mg atoms in the second layer), while the angle associated with the second layer ($\beta$) increases, eventually to 180° and becomes the next plane of the moved twin boundary (Fig. 4b). There is little change in the angle associated with the first layer ($\gamma$). This synchronized atom shuffling leads to a twin boundary migration mechanism that involves the formation of disconnections with a minimum height of two ($10\bar{1}2$) layers, consistent with the commonly accepted notion[6,32,33]. However, when Nd and Ag are present at the twin boundary (Fig. 4c), this shuffling mechanism is curtailed. As the applied shear strain increases

(Fig. 4d), the angle associated with the twin boundary ($\alpha$) remains close to 180° and obstructs the shuffling motion of the compression and expansion sites that occurs when ($\beta$) increases with applied strain in the solute-free case. While ($\alpha$) and ($\beta$) remain relatively unchanged with increasing shear strain, the angle associated with the first layer ($\gamma$) is found to increase with applied strain due to the shuffling of the site 1 Mg atom moving in the opposite direction to its two neighbouring atoms in the first layer. The angle $\gamma$ eventually reaches 180° and becomes the next plane of the moved twin boundary. This presents a new pathway for twin boundary migration via a single layer—the straightening of the first layer lowers the potential barrier that otherwise would block the slip displacement required for a single layer migration of the twin boundary. This new migration mechanism, by which the twin boundary migrates via a single layer instead of two layers (Fig. 5a), is distinctly different from that without any solute segregation.

To examine the generality of the new migration mechanism, we also computed cases where segregation in twin boundary occurs with only Nd or Ag atoms, i.e. in a binary alloy system. In the case where only Nd is present at the twin boundary (Fig. 4e), the trend of atomic shuffling is similar to that observed when Nd and Ag are both present at the twin boundary (Fig. 4d). There is little change in the angles associated with the original boundary layer ($\alpha$) and the second layer ($\beta$), only the angle of the first layer ($\gamma$) increases with an increase in the shear strain (Fig. 4f). It is conceivable that the new migration mechanism occurs in magnesium alloys containing other rare-earth elements or other specific elements, in addition to the Mg-Nd system. It is interesting to see that, in the case when only Ag is present at the twin boundary (Fig. 4g), the boundary migration mechanism is similar to that without any solute segregation (Fig. 4b). There is little change in the angle associated with the first layer ($\gamma$), while the angles of the original boundary plane and the second layer change with an increase in shear strain (Fig. 4h).

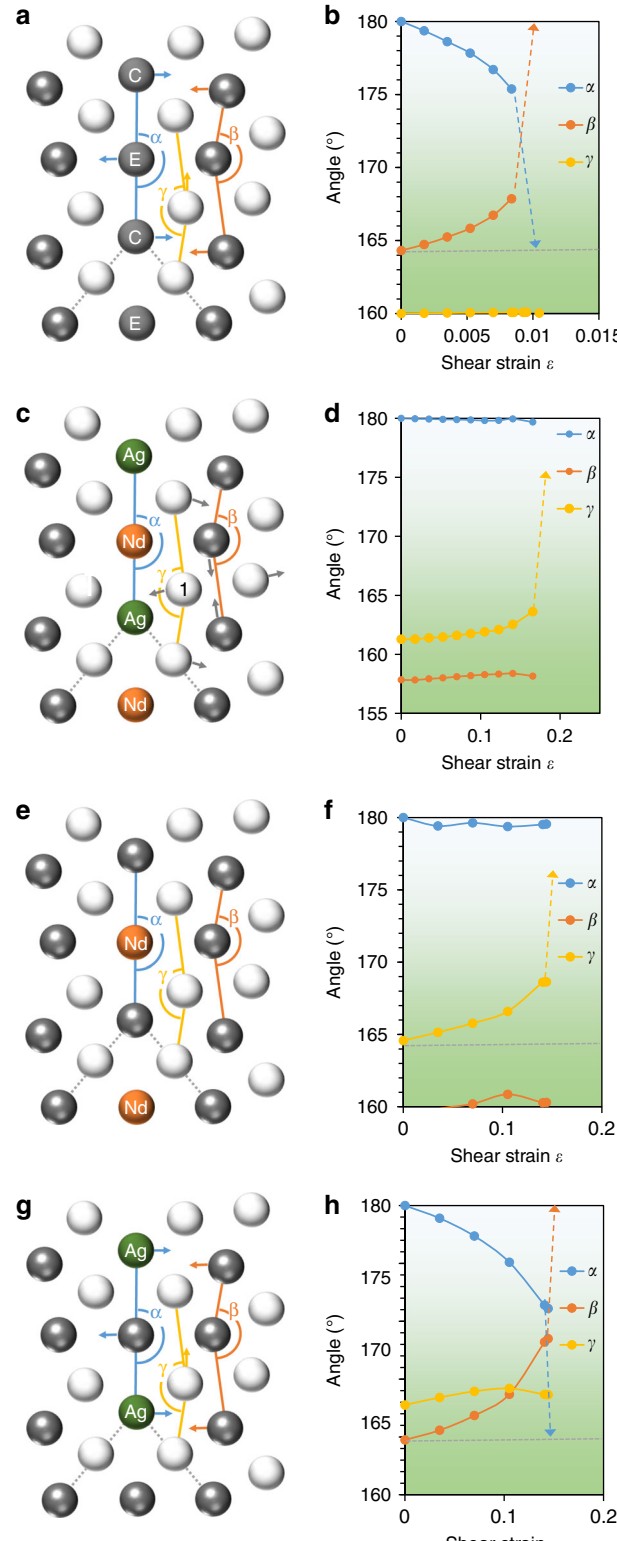

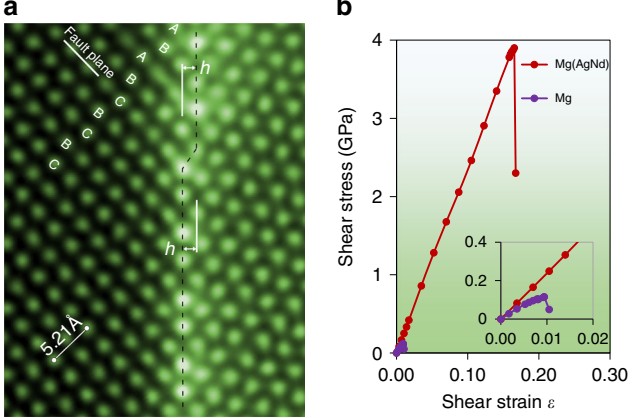

**Fig. 5** Effect of co-segregated solutes on twin boundary migration mechanism and pinning. **a** HAADF-STEM image of a (10$\bar{1}$2) twin interface containing a step of a single layer height. **b** Calculated shear stress-strain plot for the migration of a (10$\bar{1}$2) twin interface. The red line gives the shear stress when the expansion site on the twin filled with Nd and the compression site filled with Ag and the purple line is when both expansion and compression sites are filled with Mg

**Influence on twin boundary mobility**. The co-segregation of Nd and Ag atoms to twin boundary is expected to significantly reduce the twin boundary mobility. Thermodynamically, the solute segregation will reduce boundary energy and hence increase the stability and simultaneously reduce the mobility of the twin boundary. Kinetically, the solute segregation in the twin boundary will exert a pinning or drag effect on the boundary migration. The segregation of solutes to grain boundaries has been reported to generate a significant drag effect on grain boundary migration, which retards the grain growth[12,34,35]. The effect of the solute co-segregation on twin boundary mobility was examined by first-principles computation. A calculated shear stress versus strain curve for a (10$\bar{1}$2) twin boundary, with and without segregated Nd and Ag atoms, is shown in Fig. 5b. In the situation where there is no solute segregation, i.e. pure Mg, the twin boundary starts to migrate when the applied shear stress is above 116 MPa. Significant improvements in shear stress, by 33 times, and elastic strain limit occurs when the twin boundary is populated with Nd and Ag. To understand this exceptional effect of the co-segregation, a series of electron charge density plots are given in Supplementary Fig. 2. The increased charge density between Ag and Nd with the Mg indicates a stronger bond and a strengthening of the twin. As the shear deformation is applied, the Mg is pushed towards the Nd and away from the Ag.

We have demonstrated that the challenging issue of direct atomistic imaging of the structure and chemistry of interfaces in magnesium alloys and, more broadly metallic alloys, containing multiple alloying elements is now possible. With this opportunity, we have discovered an unusual segregation pattern that gives rise to a strong pinning effect on interfaces and a migration mechanism that has not previously been recognized. Our findings give insights into the precise role of segregated solute atoms in interface thermal stability and mobility that dictate mechanical behaviour and properties of many engineering materials.

**Fig. 4** Influence of segregated solute(s) on (10$\bar{1}$2) twin boundary migration mechanism. **a** Schematic of a twin boundary containing Mg at both compression (C) and expansion sites (E), and **b** variation of angles $\alpha$, $\beta$ and $\gamma$ with applied shear strain. **c** Twin boundary containing Ag at the C site and Nd at the E site, and **d** change in angles $\alpha$, $\beta$ and $\gamma$ with shear strain. **e** Twin boundary containing Nd at the E site, and **f** variation of angles $\alpha$, $\beta$ and $\gamma$ with shear strain. **g** Twin boundary containing Ag at the C site, and **h** change in angles $\alpha$, $\beta$ and $\gamma$ with shear strain

## Methods

**Sample preparation**. Cast bars of commercial alloy QE22, with a nominal composition of Mg-2.0Nd-2.5Ag-0.7Zr (wt.%), were used in the present investigation. The actual alloy composition was determined by inductively coupled plasma atomic emission spectroscopy to be Mg-2.10Nd-3.01Ag-0.34Zr (wt.%). Samples with dimensions of 9 mm × 6 mm × 6 mm were cut from the cast bar for solution treatment. The samples were covered with MgO powder, solution treated at 520 °C for 6 h, followed by water quenching. Then, the solution treated samples were used

for compression. The samples were uniaxially compressed along the longest side at room temperature, at a constant strain rate of $1 \times 10^{-3}$ to strains of 4.3% or 14.8%. Slices with thickness of 800 µm were cut from the compressed samples and then subjected to annealing at 200 °C for 5 min.

**Electron microscopy characterization.** For HAADF-STEM the samples were polished the slices to 50-70 µm, punched to discs with 3 mm in diameter, and then ion-beam milled using Gatan PIPS 695 at approximately −70 °C. Atomic-resolution HAADF-STEM images and EDS maps were acquired from a FEI Titan $G^2$ 60–300 ChemiSTEM operating at 120 kV, equipped with a Cs probe corrector and a Super-X EDS with four windowless silicon-drift detectors. A convergence semi-angle of 15 mrad, collection semi-angle of 45–262 mrad and spot size 9 were used. The count rate is in the range of 180 to 500 cps when acquiring atomic-resolution EDS maps. HAADF-STEM image simulations were conducted using xHREM software. The atomic-resolution HAADF-STEM images were Fourier filtered using Gatan DigitalMicrograph. The atomic-resolution EDS maps were appropriately adjusted for brightness, contrast and gamma value in Esprit software, without any other manipulations.

**Density functional theory (DFT) calculations.** Density functional calculations were performed using the Vienna Ab Initio Simulation Package (VASP) code[36] using the Generalised-Gradient Approximation to the DFT, with the exchange-correlation functional of Perdew-Burke-Ernzerhof[37]. The k-space integrations were performed using a Monkhorst-Pack sampling scheme[38], with an equivalent of an $(18 \times 18 \times 12)$ k-point mesh in the unit cell. Atoms were treated using the VASP provided projector augmented-wave (PAW) pseudo-potentials. For Nd, the f-electrons were frozen in the core, as this approximation has been shown to work well with RE-doped Mg[39]. Geometry optimizations were performed using a conjugate gradient algorithm, with relaxations terminated when the forces on all atoms decreased below 0.008 eV/Å. Models of the coherent twin boundaries were formed using 80 atoms for the $(10\bar{1}1)$ coherent twin boundary and 68 atoms for the $(10\bar{1}2)$ coherent twin boundary. The shear stress versus strain was calculated by straining the supercell in a series of incremental shears, relaxing the atomic positions at each step. To create a continuous strain path, the relaxed coordination from the previous step was used as the starting positions of the next.

## Data availability

All relevant data are available from the corresponding author on request.

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

## Acknowledgements

The authors are grateful for financial support from the National Natural Science Foundation of China (51771036, 51131009 and 51421001), National Key Research and Development Program of China (2016YFB0700402), and the Australian Research Council. This work is supported by the resources provided by the National Computational Infrastructure National Facility (NCI–NF).

## Author contributions

J.F.N. designed the experiments, analysed the data, interpreted the results and wrote the paper. X.J.Z. carried out HAADF-STEM and STEM-EDS experiments. H.W.C. did HAADF-STEM simulation. N.W. carried out the DFT calculations and their analysis. H.W.C. and Q.L. participated the project and provided discussion and proof-reading of the manuscript.

## Additional information

**Competing interests:** The authors declare no competing interests.

