## [Transparent Peer Review File · Nature Communications]

Reviewers' comments:

Reviewer #1 (Remarks to the Author):

The paper deals with one of the most important issues in Mg alloys; segregation of solute atoms along twin boundaries. Utilizing the atomic-resolution energy dispersive X-ray spectroscopy at a low electron voltage, the authors have done an excellent microscopy work, which shows that Nd and Ag atoms co-segregate, forming alternating columns that fully occupy the twin boundary. However, the paper, as-is, lacks the scientific merit that is expected for the paper published in Nature Communications.

Although the authors have tried to correlate such phenomenon with the migration of twin boundary (or pinning effects of segregation), it actually does not provide any answers to the important question; the effect of segregation on microstructural and texture evolution. It is highly desired that the authors mention about the issue.

It is known that in the case of Mg-RE(Ca)-Zn alloy system, RE(Ca) atoms larger than Mg atoms segregate to the region of the missing half-plane below the slip plane and Zn atoms smaller than Mg atoms to the region of the extra half-plane above the slip plane. On the other hand, in the case of Mg-Gd-Zn alloy, both Gd and Zn atoms segregate to the region of the missing half-plane below the slip plane. However, in the present result of Mg-Nd-Ag alloy, it is shown that Nd atoms segregate exclusively to the region of the missing half-plane below the slip plane and Ag atoms to the region of the extra half-plane above the slip plane, although both Nd and Ag atoms are larger than Mg atoms. Can the authors comment on this rather contradicting result depending on alloy systems?

Reviewer #2 (Remarks to the Author):

Overall, the article is short and clearly written and easy to follow. In this study, the authors characterize the structure and chemistry of two types of twin boundaries in a deformed Mg alloy (containing Nd and Ag). The experimental characterization is conducted at the atomic scale, requiring a sophisticated, state-of-the-art capability, namely, an atomic-resolution EDS used at a much lower electron voltage than standard. As they describe well, these experiments have been performed with great care. Their main finding is that the larger solute atom (in this case Nd) segregates to the extension sites in the twin boundary and the smaller solute atom (in this case Ag) to the compression sites in the (10-12) twin boundary. They show that the Nd and Ag co-segregation pattern also occurs in another type of twin boundary, the (10-11) boundary, in the same Mg alloy, also possessing extension and compression sites. DFT calculations are performed to provide support for co-segregation, showing that these particular Nd and Ag solute positions within the twin boundary are more favorable than when positioned in the bulk crystal. A DFT model of the decorated twin boundary is then developed to show that the twin boundary is more resistant to shear than the undecorated twin boundary. Last, changes in twin boundary angle are analyzed to suggest how the Nd and Ag-filled boundary would influence boundary migration.

The authors claim two original aspects of this work. The first is the technique. The atomic-scale EDS technique is impressive and important, and as they clearly demonstrate here, using a lower voltage enables characterization of solute type and arrangement within boundaries. In this paper, they provide a short demonstration of this technique; they apply it to one material and to ideal boundaries, which are coherent (twins), within this material. The second claim is the discovery of co-segregation. In prior work, the authors showed that two types of solutes can also segregate to the twin boundary, although segregation of the two solutes was to two extension sites, and not to the extension/compression sites seen here for the Ag and Nd solute elements. This is an

interesting observation but it could be better motivated or reasoned. Was there a goal in mind for this study? For instance, no explanation for why this segregation behavior would be different. Why was this alloy selected? As far as segregation of the relatively smaller solute Ag and relative larger one, Nd, would they have expected something else (no segregation or just one solute to segregate and not the other)?

To conclude, while the work is important, in the reviewer's view the results are not so surprising as the authors may think. The segregation of smaller atoms and larger atoms to these compression and extension sites, respectively, in the twin boundary, can be expected to be favorable to relieve the local stresses imparted by the defect (the twin boundary). The lower boundary energy they calculate for the Ag-Nd twin boundary would naturally follow, as well as the increase in shear stress. Also, the authors conduct a post-mortem characterization and static DFT calculations, yet there are frequent claims made on kinetic processes, such as boundary migration and co-segregation, i.e., they conclude "we have discovered an unusual segregation pattern that gives rise to an exceptionally strong pinning effect on interfaces and a migration mechanism that has not previously been recognized." This statement is misleading; while they demonstrate that Nd and Ag are attracted to extension and compression sites in twin boundaries, they have not provided enough support to show the strong pinning effect or a migration mechanism. In summary, this work shows an interesting result and it should be published but it is not suitable for Nat Communications.

As minor comments, there are several places where statements are made that are in need of more explanation. Such as the claim that Nd is more stable than Ag under the electron radiation. Stable in what way? What behavior was observed that would indicate more stability? Also, the final section on the connection between twin boundary angle and migration is not clearly explained or motivated.

Reviewer #3 (Remarks to the Author):

Manuscript by Zhao, Chen et al

This manuscript presents very nice work on atomic-resolution chemical analysis on grain boundaries in Mg-based alloys. While there has been work on imaging of atoms at interfaces and grain boundaries in oxides in the past (from other groups), this is, to the best of my knowledge the first EDS on multiple alloys in Mg. What is more significant is that the work is also going in depth with respect to the interpretation of the data, comparing the experimental results related to segregation with energetics calculations demonstrating why and supporting the experimental results. Also, the authors went further by deducing how the results affect the mechanical behaviour. I consider this would be a very valuable paper for the field of alloys. In Summary, the work is convincing and of interest to the community.

I do have a question related to the images, which should be considered for completeness in analysis. The EDS results show the Ag and Nd nicely but the HAADF images (Fig 1a and b) show that the higher intensity of the atomic columns goes well beyond the single atomic plane. How can the authors reconcile this local (single plane) interpretation with the broader width of the contrast? On Figure 1 f one also sees broader bands away from the boundary (7-8 atoms away from the boundary, then again around 20 planes away) on the upper grain and a darker band nearer the boundary. We also see very fine (1 plane) bright lines within the "lower" grain from the boundary. Are there also other types of segregation mechanisms or correlations in solid solution? What are these effects due to? This could be important if there are solutes in the matrix, affecting the mechanical properties.

Also, how about the contrast in Figure 4f where the column intensity of the atoms on near the

boundary (on the left side of the boundary) near the shuffle is higher as compared to the bottom of the field of view? Is there a 3D effect or more complex segregation structure?

Responses to Reviewers' Comments

We would like to thank the reviewers and the editor for their constructive comments and instructions on our paper. We have revised our manuscript accordingly, including the addition of 10 extra references, for clarity. All changes are highlighted in the revision. We hope that the revised version is now ready for acceptance for publication.

We would also like to take this opportunity to emphasize the three novelties in this paper:

1. Direct observation and identification of multiple alloying elements that segregated at the atomic scale in a magnesium alloy. It represents the first atomic resolution EDS maps of solute segregation in metallic materials. The segregation in these materials is prone to electron beam damage, but we demonstrate that it is possible to solve this difficulty by doing atomic-resolution EDS at a much lower electron voltage.
2. A new pattern of solute segregation: alloying elements co-segregate, forming alternating columns that fully occupy the twin boundary, Fig. R1(a), in contrast to the previous observations of half occupancy of the boundary where mixed-solute columns alternate with magnesium, Fig. R1(b). Each of the two patterns possibly represents the solute segregation in a group of alloys having complex compositions.
3. A new mechanism of twin boundary migration: the segregation of specific solutes (either in combination or individually) switches the migration mechanism of the twin boundary from the commonly accepted mode (nucleation and lateral sliding of ledges or disconnections that have a minimum height of two layers of the boundary plane) to a new one (nucleation and lateral sliding of ledges or disconnection that have a minimum height of one layer of the boundary plane), which is now elaborated in detail in Figure 4 and associated text in the revision.

Figure R1. (a) Co-segregation in a twin boundary, with alternating columns of larger and smaller solute atoms, (b) Co-segregation within single columns in a twin boundary, with alternating columns of Mg and solute (larger and smaller solutes in a single column).

Reviewer #1

- 1 “Although the authors have tried to correlate such phenomenon with the migration of twin boundary (or pinning effects of segregation), it actually does not provide any answers to the important question; the effect of segregation on microstructural and texture evolution. It is highly desired that the authors mention about the issue.”

We have added the following in the revision: “It is now known that the addition of RE elements to magnesium alloys can significantly weaken the recrystallization texture, and that a combined addition of RE and non-RE elements may generate an even weaker recrystallization texture than the single addition of RE^{10, 11}. The RE additions lead to more deformation twins that

provide more nucleation sites for recrystallization grains having random orientations. It has been reported recently^{12, 13} that the combination of larger and smaller atoms of appropriate alloying elements can lead to much weaker texture and better formability by maximizing the co-segregation of these texture-controlling elements in grain boundaries.”.

- 2 “It is known that in the case of Mg-RE(Ca)-Zn alloy system, RE(Ca) atoms larger than Mg atoms segregate to the region of the missing half-plane below the slip plane and Zn atoms smaller than Mg atoms to the region of the extra half-plane above the slip plane. On the other hand, in the case of Mg-Gd-Zn alloy, both Gd and Zn atoms segregate to the region of the missing half-plane below the slip plane. However, in the present result of Mg-Nd-Ag alloy, it is shown that Nd atoms segregate exclusively to the region of the missing half-plane below the slip plane and Ag atoms to the region of the extra half-plane above the slip plane, although both Nd and Ag atoms are larger than Mg atoms. Can the authors comment on this rather contradicting result depending on alloy systems?”

Nd and Ag atoms are NOT both larger than Mg atoms. The atomic metallic radius is 1.81 for Nd, 1.44 for Ag, and 1.60 for Mg. We think that the reviewer refers to solute segregation to the core of an edge dislocation, rather than in a twin boundary. In the case a coherent twin boundary that is covered in our paper, there are NO dislocations in the boundary. Note that we did not show “Nd atoms segregate exclusively to the region of the missing half-plane below the slip plane and Ag atoms to the region of the extra half-plane above the slip plane”. What we show and report in our paper is a new pattern of solute segregation: segregated solutes form alternating columns that fully occupy the twin boundary, Fig. R1(a), in contrast to the previous observations of half occupancy of the boundary where mixed-solute columns alternate with magnesium, Fig. R1(b).

Reviewer #2

1. “The authors claim two original aspects of this work.”

There are three novelties in this paper, as we stated in the previous page of our Response Letter. Apart from the two aspects mentioned by the reviewer, the third novelty is a new mechanism of twin boundary migration: the segregation of specific solutes (either in combination or individually) switches the migration mechanism of the twin boundary from the currently accepted mode to a new one, as elaborated in Figure 4 and associated text in the revision.

2. “The second claim is the discovery of co-segregation. In prior work, the authors showed that two types of solutes can also segregate to the twin boundary, although segregation of the two solutes was to two extension sites, and not to the extension/compression sites seen here for the Ag and Nd solute elements. This is an interesting observation but it could be better motivated or reasoned. Was there a goal in mind for this study? For instance, no explanation for why this segregation behavior would be different. Why was this alloy selected? As far as segregation of the relatively smaller solute Ag and relative larger one, Nd, would they have expected something else (no segregation or just one solute to segregate and not the other)?”

We selected Nd because (i) it is one of the major alloying elements in Mg alloys (ii) other rare-earth elements are also commonly used as alloying elements to Mg, all of which have larger atomic size than Mg. Ag is also one of the most effective alloying element in Mg alloys (e.g. commercial alloy EQ22, Mg-Gd-Ag and Mg-Y-Ag experimental alloys). We have added the following on page 2 of the revision: “The alloy system that we selected in this work, Mg-RE-Ag (where RE represents rare-earth), forms the base of a group of magnesium alloys that have superior mechanical properties at both ambient and elevated temperatures^{30, 31}. The alloy

contains Nd and Ag. Nd has a larger atomic size than Mg, Ag has a smaller size than Mg. Having higher atomic numbers on the periodic table, Nd and Ag themselves would have been unsuitable for Z-contrast imaging. Their distribution at the atomic scale can be revealed only by EDS.”

The previously observed pattern of solute segregation in twin boundaries [Reference 3 in the revision] is that columns containing both larger and smaller solutes alternate with those of Mg in the twin boundary, occupying half of the boundary, as shown clearly in Fig. R1(b). What was reported in this paper is a surprising observation: alloying elements co-segregate, forming alternating columns that fully occupy the twin boundary, see Fig. R1(a) and other figures in our paper. This newly discovered pattern of solute segregation was fully explained based on our DFT calculation results. As shown in Figure 3 in the revised manuscript, comparison of the two possible segregation patterns outlined in Fig. R1, indicates that Nd and Ag atoms prefer to co-segregate forming alternating columns that fully occupy the twin boundary, Fig. R1(a). The newly discovered pattern possibly represents the solute segregation in a group of alloys having complex compositions, opening the door for more research in the future. We have modified the text (last paragraph of page 3 of the revision) to distinguish the solute segregation pattern reported in this work and the solute segregation pattern reported previously.

3. “To conclude, while the work is important, in the reviewer’s view the results are not so surprising as the authors may think. The segregation of smaller atoms and larger atoms to these compression and extension sites, respectively, in the twin boundary, can be expected to be favorable to relieve the local stresses imparted by the defect (the twin boundary). The lower boundary energy they calculate for the Ag-Nd twin boundary would naturally follow, as well as the increase in shear stress. Also, the authors conduct a post-mortem characterization and static DFT calculations, yet there are frequent claims made on kinetic processes, such as boundary migration and co-segregation, i.e., they conclude “we have discovered an unusual segregation pattern that gives rise to an exceptionally strong pinning effect on interfaces and a migration mechanism that has not previously been recognized.” This statement is misleading; while they demonstrate that Nd and Ag are attracted to extension and compression sites in twin boundaries, they have not provided enough support to show the strong pinning effect or a migration mechanism.”

In prior work in the Mg-Gd-Zn ternary system [Reference 3 in the revision], both the smaller (Zn) and larger (Gd) cations were found to locate in the expansion sites; the segregation of smaller Zn atoms into Gd columns led to a greater reduction in system energy than forming alternating columns of Zn and Gd. The co-segregation of smaller (Zn) and larger (Gd) solutes in a single column is driven by the minimization strain along the column direction that would otherwise be created by exclusive occupancy of larger atoms. The segregation of smaller atoms to the compression site and larger atoms to the expansion site would not seem the obvious way for the system to relieve local stress; this is not always the case.

It was not our intention to make claims on any kinetic processes. In the revision, we have now removed those words that may cause confusion. What we reported is purely on the change of migration mechanism of a twin boundary in the presence of solutes in the boundary. We have not modelled kinetic processes in the boundary migration, rather we have used a set of static DFT calculations to understand how the geometry of the boundary changes under strain in the lead up to migration. There are clear differences in the way the atoms around the twin boundary respond to shear, depending upon the incorporation of solutes. Under strain the atomic shuffling of the first two Mg layers occurs in such a way as to encourage a twin boundary migration mechanism that involves two $(10\bar{1}2)$ layers, consistent with the commonly accepted notion [References 6, 32 and 33 and page 4 of the revision], Fig. 4 in the revision. Whereas with the incorporation of Nd and Ag, or Nd alone, our static calculations show that the first

layer straightens under strain, which would lower the energy for a single layer migration of the twin boundary. While modelling the kinetics of the twin migration is beyond the scope of this work, we included experimental evidence of the single layer steps occurring in a Mg-Nd-Ag ternary alloy to support this conclusion. In the revision, we have included more DFT results to show the effect of specific solute(s) on twin boundary migration mechanism, which has not been explored in the past.

Regarding solute pinning effects, we have included the following in the revision (top paragraph on page 5): “The co-segregation of Nd and Ag atoms to twin boundary is expected to significantly reduce the twin boundary mobility. Thermodynamically, the solute segregation will reduce boundary energy and hence increase the stability and simultaneously reduce the mobility of the twin boundary. Kinetically, the solute segregation in the twin boundary will exert a pinning or drag effect on the boundary migration. The segregation of solutes to grain boundaries has been reported to generate a significant drag effect on grain boundary migration, which retards the grain growth^{12, 34, 35}”.

4. “As minor comments, there are several places where statements are made that are in need of more explanation. Such as the claim that Nd is more stable than Ag under the electron radiation. Stable in what way? What behavior was observed that would indicate more stability?”

We mentioned in the original manuscript that Nd-rich columns are more stable than Ag-rich columns under continuous electron radiation. We observed this phenomenon when we were acquiring EDS maps. To obtain an unambiguous high-resolution EDS map with enough signals, we needed to control the convergent electron beam scanning within a tiny area (around $2 \text{ nm} \times 2 \text{ nm}$) of the sample for about 100 seconds. When acquiring an EDS map along the $[\bar{1}2\bar{1}0]$ direction, where the Nd-rich and Ag-rich columns alternately distributed in the twin boundary (Figs. 1a-1e and Figs. 2a-2e), the intensity of the Nd-rich and Ag-rich columns in the corresponding HAADF-STEM image gradually decreased under the continuous scanning of electron beam. This led us to think that the Nd-rich and Ag-rich columns are not stable under the continuous electron radiation. Furthermore, we also found that the intensity of the Nd-rich columns decreased much more slowly than that of Ag-rich columns. In other words, Nd-rich columns can stay much longer than Ag-rich columns under identical dose of electron radiation. To illustrate this point, Figure R2 shows a series of HAADF-STEM images of a section of a coherent $(10\bar{1}2)$ twin boundary which has been exposed for different time to electron radiation (continuous scanning by a convergent electron beam operating at 300 kV). It is seen clearly that, after around 50 seconds electron radiation under 300 kV, Ag-rich columns become much darker, while there is little change in the intensity of Nd-rich columns. It is for this reason that, in the present work, we lowered the accelerating voltage of electrons from 300 to 120 kV to reduce the radiation damage (especially to Ag-rich columns) and successfully acquired enough signals for Ag in the Ag-rich columns before these Ag-rich columns were fully damaged by the electron radiation.

In the revision, we have now added an extra Extended Data figure (Extended Data Fig. 1) to demonstrate that the Nd-rich columns are more stable than Ag-rich columns under continuous electron radiation.

Figure R2. A series of HAADF-STEM images showing the stability of Nd-rich and Ag-rich columns in a coherent $(10\bar{1}2)$ twin boundary under continuous electron radiation for different scanning times (given in each image). Electron beam is parallel to the $[\bar{1}2\bar{1}0]$ direction. Accelerating voltage of electrons is 300 kV.

5. “Also, the final section on the connection between twin boundary angle and migration is not clearly explained or motivated.”

We first apologize that the labelling in original Fig. 4 mislabelled the figure parts that may have made the discussion on twin boundary geometries more difficult to read. The motivation to look at the twin boundary geometry was to understand how the twin boundary responded to applied external strain. The strained twin boundary geometries are the starting point of twin migration and so play a profound role in these processes. In the revision, we have now provided some detailed explanation of the connection between twin boundary angle and migration mechanism.

Reviewer #3

1. “I do have a question related to the images, which should be considered for completeness in analysis. The EDS results show the Ag and Nd nicely but the HAADF images (Fig 1a and b) show that the higher intensity of the atomic columns goes well beyond the single atomic plane. How can the authors reconcile this local (single plane) interpretation with the broader width of the contrast? On Figure 1f one also sees broader bands away from the boundary (7-8 atoms away from the boundary, then again around 20 planes away) on the upper grain and a darker band nearer the boundary. We also see very fine (1 plane) bright lines within the “lower” grain from the boundary. Are there also other types of segregation mechanisms or correlations in solid solution? What are these effects due to? This could be important if there are solutes in the matrix, affecting the mechanical properties.”

We think the extra contrast near the single atomic plane is due to the presence of atomic-scale steps along the viewing direction. Figure R3 shows steps distributed along $[\bar{1}011]$ and $[\bar{1}2\bar{1}0]$, two orthogonal directions. In these and other images that we collected, the height of the steps varies from 1 to 12 interplanar spacing of $(10\bar{1}2)$, while the length of each coherent twin boundary, i.e. the section between two neighbouring steps, is in the range 5-40 nm. Since the foil thickness is usually around 30-60 nm, it is possible that the observed region contains several steps along the viewing direction, as shown schematically in Fig. R4. To check the validity of this speculation, we took a set of through-focus HAADF-STEM images, Fig. R5, using the twin boundary in Fig. 1f (in original manuscript) as an example. This boundary was selected because the influence of the steps along the viewing direction is most severe. As

shown in Fig. R4, three fully coherent sections of the twin boundary could be focused with different defocus (Δf). The focused coherent twin boundary exhibits a single atomic plane with higher intensity, while the other two unfocused coherent twin boundaries look like broader bands with a little higher intensity than surrounding matrix. For the probe semi-angle of 15 mrad that we used for imaging in the present study, the depth of focus is ~ 15 nm.

As for the coherent twin boundary (CTB) shown in Fig. 1a in the original manuscript, it was part of a boundary shown in Fig. R6a. Note that there is a step close to the region of Fig. 1a. The two adjacent CTBs associated with the step have a transition zone. Such steps were frequently observed in the present study. We speculate that the broader width of the contrast of the CTB in Fig. 1a is mainly due to the presence of the adjacent CTB. A HAADF image showing a segregated CTB with less influence by its adjacent CTB is shown in Fig. R6(b). The influence is almost invisible in the bottom-right corner.

Figure R3. HAADF-STEM images showing $(10\bar{1}2)$ twin boundaries with steps distributing along (a) $[\bar{1}011]$ and (b) $[\bar{1}2\bar{1}0]$ directions. Electron beam is parallel to $[\bar{1}2\bar{1}0]$ in (a) and $[\bar{1}011]$ in (b).

Figure R4. Schematic diagram showing the presence of steps in a $(10\bar{1}2)$ twin boundary.

Figure R5. Through-focus HAADF-STEM images showing a $(10\bar{1}2)$ twin boundary having three coherent terraces lying parallel to the electron beam direction. These three coherent terraces are separated by steps along the electron beam direction. Electron beam is parallel to $[\bar{1}011]$.

Figure R6. (a) HAADF-STEM image showing a $(10\bar{1}2)$ twin boundary. The section of the boundary within the green frame was used for Fig. 1a in the original manuscript. The contrast of this section is influenced by its adjacent coherent section (arrowed). (b) HAADF-STEM image showing a coherent twin boundary that is less influenced by its adjacent coherent twin boundary, especially in the region that is remote from the step, i.e. the bottom-right corner.

2. “Also, how about the contrast in Figure 4f where the column intensity of the atoms on near the boundary (on the left side of the boundary) near the shuffle is higher as compared to the bottom of the field of view? Is there a 3D effect or more complex segregation structure?”

This part is more complex than what we could unambiguously explain. One possibility is that this contrast is generated by the 3D effect that we described before, i.e. there is possibly a step in the boundary through the specimen thickness in this area. The existence of the 3D effect makes it difficult for us to explain the origin of the contrast here. We would prefer to leave this aspect for future investigation.

REVIEWERS' COMMENTS:

Reviewer #1 (Remarks to the Author):

The revised version is suitable for publication in Nature Communications.

Reviewer #2 (Remarks to the Author):

The authors have adequately addressed the concerns of this reviewer and the revised manuscript is recommended for publication.

Reviewer #3 (Remarks to the Author):

I am fine with the changes related to my original review.